# Quinolizidines as Novel SARS-CoV-2 Entry Inhibitors

**DOI:** 10.3390/ijms23179659

**Published:** 2022-08-25

**Authors:** Li Huang, Lei Zhu, Hua Xie, Jeffery Shawn Goodwin, Tanu Rana, Lan Xie, Chin-Ho Chen

**Affiliations:** 1Department of Surgery, Duke University Medical Center, Durham, NC 27710, USA; 2School of Dentistry, Meharry Medical College, Nashville, TN 37208, USA; 3Department of Biochemistry and Cancer Biology, Meharry Medical College, Nashville, TN 37208, USA; 4Eshelman School of Pharmacy, University of North Carolina at Chapel Hill, Chapel Hill, NC 27599, USA

**Keywords:** SARS-CoV-2 inhibitor, aloperine, aloperine derivatives

## Abstract

COVID-19, caused by the highly transmissible severe acute respiratory syndrome coronavirus-2 (SARS-CoV-2), has rapidly spread and become a pandemic since its outbreak in 2019. We have previously discovered that aloperine is a new privileged scaffold that can be modified to become a specific antiviral compound with markedly improved potency against different viruses, such as the influenza virus. In this study, we have identified a collection of aloperine derivatives that can inhibit the entry of SARS-CoV-2 into host cells. Compound **5** is the most potent tested aloperine derivative that inhibited the entry of SARS-CoV-2 (D614G variant) spike protein-pseudotyped virus with an IC_50_ of 0.5 µM. The compound was also active against several other SARS-CoV-2 variants including Delta and Omicron. Results of a confocal microscopy study suggest that compound **5** inhibited the viral entry before fusion to the cell or endosomal membrane. The results are consistent with the notion that aloperine is a privileged scaffold that can be used to develop potent anti-SARS-CoV-2 entry inhibitors.

## 1. Introduction

Since its outbreak in 2019, Coronavirus Disease 2019 (COVID-19) has rapidly spread and become a pandemic [1]. Based on current information from World Health Organization (WHO), there have been more than 500 million confirmed COVID-19 cases, with more than 6 million deaths globally as of June 2022. COVID-19 is caused by the highly transmissible severe acute respiratory syndrome coronavirus-2 (SARS-CoV-2) [2,3,4,5]. SARS-CoV-2 may cause severe illnesses to the respiratory system and, in some cases, other organs. Many potential clinical remedies were tested for their efficacy against SARS-CoV-2, including chloroquine, hydroxychloroquine, lopinavir plus Ritonavir (Kaletra), umifenovir (Arbidol), remdesivir (RE), and favipiravir [6]. However, most of the repurposing drugs did not present significant clinical improvement in hospitalized adult COVID-19 patients [7]. Recently, two orally effective drugs Paxlovid and Molnupiravir were approved by the US FDA for emergency use of COVID-19 treatment. Paxlovid has two drug components—one is Nirmatrelvir, a peptidomimetic inhibitor targeting the SARS coronavirus main protease (Mpro) to block viral polyprotein processing during viral replication, and the other is Ritonavir that inhibits the metabolic break-down of Nirmatrelvir to prolong efficacy [8,9]. Molnupiravir is a nucleoside analog, which interferes with viral RNA transcription by targeting viral RNA-dependent RNA polymerase (RdRp) [8]. However, due to the high mutation rate of SARS-CoV-2, new variants continue to emerge that may compromise the effectiveness of current vaccines and drugs. The D614G was the first to replace the original SARS-CoV-2 as globally dominant variant due to its increased receptor binding and better fitness [10]. The WHO has since named multiple variants of concern (VOC) including Alpha, Beta, Gamma, Delta, and the recently circulating Omicron variants. It was reported that Omicron variants contain more mutations than previous variants, especially in the receptor-binding domain of the viral spike protein, which could lead to resistance to neutralizing antibodies. It was reported that Omicron variants are much more infectious compared to previous prevalent SARS-CoV-2 subtypes such as Delta, while partially resistant to the vaccine-induced neutralizing antibodies [11,12,13]. Based on these information, it is highly possible that new variants resistant to current vaccines and treatments could emerge in the future. Thus, novel anti-SARS-CoV-2 agents that can inhibit a broad spectrum of SARS-CoV-2 variants are urgently needed.

The life cycle of SARS-CoV-2 presents various opportunities for the development of novel anti-SARS-CoV-2 therapeutics. This study was focused on identifying novel small molecules that can effectively inhibit SARS-CoV-2 entry. The virus uses ACE2 as a receptor to enter cells through two routes: endocytosis and direct fusion with the cell membrane [14]. Many viruses, including SARS-CoV-2 and influenza viruses, use endocytosis to enter host cells [15,16]. The endosomal cathepsin B may be responsible for cleavage of the viral spike protein (S), which results in membrane fusion and release of the viral RNA into cytoplasm. The second route of viral entry, through the cell membrane, requires cellular proteases, such as the transmembrane protease serine 2 (TMPRSS2), which cleaves the S protein into S1 and S2 subunits for membrane fusion [17,18]. Once it enters into the host cell cytosol, the viral genomic RNA is directly translated by host ribosomes in the cytoplasm to complete viral replication. Each step of the viral entry process may be a potential target for therapeutic intervention.

Our rationale for looking into the anti-SARS-CoV-2 activity of aloperine and its derivatives stems from our prior studies on the antiviral activities of this class of compounds. We have previously discovered that aloperine is a new privileged scaffold that can be modified to become a specific antiviral compound with markedly improved potency against different viruses such as the influenza virus or HIV-1 [19,20,21,22]. Due to their potential of having a broad spectrum antiviral activity, we tested aloperine and a series of previously reported aloperine derivatives against a SARS-CoV-2 pseudovirus containing the corona virus spike protein of the D614G variant [23]. The pseudoviruses also contain a luciferase gene as a reporter, which can be used to efficiently screen small molecules or antibodies that can block the virus entry. Since SARS-CoV-2 has evolved into multiple variants with significant mutations in the spike proteins, it is important to test compounds against variants especially the recently circulating Omicron variants. The significance of this study includes identification of aloperine derivatives with much improved anti-SARS-CoV-2 entry activity through structural modifications and demonstration of their ability to inhibit the pseudotyped viruses carrying SARS-CoV-2 spike proteins from various variants, such as that from currently circulating Omicron BA.4/BA.5. The results of this study described herein are expected to provide critical information towards further developing this class of natural products into effective therapeutics for the treatment of COVID-19.

## 2. Results

### 2.1. Anti-SARS-CoV-2 Entry Activity

Aloperine was active against both influenza virus (PA/Puerto Rico/8/1934, PR8) and HIV-I (NL4-3 nano-luc) at an IC_50_ of 14.5 and 1.75 µM, respectively [19,20,21,22]. In this study, aloperine exhibited moderate activity against D614G spike-pseudotyped virus entry with an IC_50_ of 11.5 µM without affecting cell viability (Table 1). Aloperine derivatives **1**–**8** with a range of structural diversity, especially at the N12 side chains, were also tested in the same assay using the D614G spike-pseudotyped reporter virus.

The aloperine derivatives **1**–**8** exhibited a range of activity against D614G spike-pseudotyped virus from inactive to sub-µM inhibition (Table 1). Compound **1** was moderately active against the D614G spike-pseudotyped virus infection with an IC_50_ of 4.7 µM. In contrast to compound **1**, compound **3** potently inhibited HIV-1 at an IC_50_ of 0.12 µM without anti-influenza virus activity. Similar to compound **1**, compound **3** was moderately active against the D614G spike-pseudotyped virus infection with an IC_50_ of 3.8 µM. Compound **7** was previously found to be equally active against both the HIV-1_NL4-3_ and the influenza A virus PR8 with an IC_50_ of 0.80 and 0.83 µM, respectively [22]. The potency of compound **7** against the D614G spike-pseudotyped virus was comparable to that of compounds **1** and **3** with an IC_50_ of 3.7 µM. These results suggest that the aloperine derivatives have distinct structure–activity relationships (SARs) when compared to that of their anti-HIV-1 or anti-influenza virus activities. Thus, the anti-SARS-CoV-2 entry activity of aloperine derivatives cannot be predicted from their anti-HIV-1 or anti-influenza virus activity.

Among the tested aloperine derivatives, compound **5** exhibited the most potent activity against the D614G spike-pseudotyped virus infection with an IC_50_ of 0.50 µM (Table 1), which was approximately 22- and 5-fold more potent than aloperine and chloroquine, respectively. Compound **5** exhibited anti-HIV activity at an IC_50_ of 0.96 µM, but it was ineffective against the influenza A virus PR8. In contrast, compound **5** was inactive against a murine leukemia virus envelope (MLV-env) pseudotyped virus where the SARS-CoV-2 spike protein was replaced with MLV-env. Compound **8** was inactive for both HIV-1 and influenza A virus and was a weak inhibitor for D614G spike-pseudotyped virus. The data also support the notion that the SARs of aloperine derivatives against SARS-CoV-2 is distinct from that of their anti-HIV or anti-influenza A virus activities.

Compounds **4** and **5** were significantly more potent than other structurally similar aloperine analogs for inhibition of the D614G spike-pseudotyped virus infection. Both compounds exhibited sub-µM potency against the D614G spike-pseudotyped virus infection of 293T-ACE2 cells. Compounds **4** and **5** are structurally different from other less potent analogs in that they possess an amine instead of an amide moiety to connect the terminal aromatic group to the aliphatic linker. Thus, we further synthesized compound **9** with the same amine moiety that has a fluorine in the para position of the aromatic ring using a similar method for obtaining compound **5** [20]. The 1H and 13C NMR information of compound **9** were included in Appendix A (see Appendix A). Compound **9** exhibited comparable anti-D614G spike-pseudotyped virus activity when compared with compound **5** (Table 1), suggesting that the amine moiety in the N12 side chain is favored for potent anti-SARS-CoV-2 activity.

### 2.2. Effect of Compound **5** on SARS-CoV-2 Variants

The emergence of new SARS-CoV-2 variants such as Delta, Omicron BA.1, BA.2, and BA.4/BA.5 has caused new waves of infection around the world. To determine the effectiveness of the aloperine derivatives on various SARS-CoV-2 variants, we tested the ability of the compounds to block entry of D614G, Delta, Omicron BA.1, Omicron BA.2, and Omicron BA.4/BA.5 with the same assay system described above. The results summarized in Table 2 indicated that D614G and Delta pseudotyped viruses were approximately equally sensitive to compound **5**, whereas all the tested Omicron variants were approximately 1.5-fold less sensitive to the compound. In contrast, potency of the cathepsin inhibitor E64D was increased against the Omicron spike-pseudotyped viruses. On the other hand, the Omicron variants were very resistant to the spike protein mAb DH1047 [24]. These results suggest that, although slightly decrease in potency, the aloperine derivative **5** remains active against all tested variants, including the currently circulating Omicron variants.

### 2.3. Mechanism of Action Study

The D614G spike-pseudotyped virus is an indicator virus for the SARS-CoV-2 spike protein-mediated cell entry [23]. Inhibition of the D614G spike-pseudotyped virus infection suggests that the aloperine derivatives blocked the virus from entering the cells. Previous reports suggested that SARS-CoV-2 may enter cells through direct fusion with the cell membrane or endosomal membrane after endocytosis [14]. To dissect the mechanism of the anti-SARS-CoV-2 entry activity of aloperine derivatives, we infected 293T-ACE2 cells with the D614G spike-pseudotyped virus in the presence or absence of compound **5** or chloroquine diphosphate for 2 h. The SARS-CoV-2 spike protein was then detected with a fluorescence-conjugated anti-spike protein antibody and visualized under a confocal microscope using a protocol we have described previously [19]. As shown in Figure 1A, the spike protein with green fluorescence was barely visualized in the absence of compound **5**, likely due to their low abundance and/or degradation by cellular proteolytic activities quickly after infection. In contrast, the virus was observed as puncta in 293T-ACE2 cells in the presence of compound **5** (Figure 1B), suggesting that the virus was arrested on the cell membrane or in endosome in the presence of the compound **5**. The presence of the spike proteins in the 293T-ACE2 cells raised the possibility that compound **5** did not block the binding of the pseudotyped virus to the ACE2 receptor. In addition, chloroquine treated sample showed no accumulation of viral puncta (Figure 1C), which suggested a difference in their mechanisms of action for chloroquine and compound **5**. Chloroquine was reported to inhibit SARS-CoV-2 at various steps of the viral life cycle [25].

A class of aloperine derivatives was reported to have moderate anti-SARS-CoV-2 activity [26]. The highlighted compound **8a** in the mentioned report exhibited an IC_50_ of 19 µM, while compound **5** described herein had an IC_50_ of 0.5 µM using a comparable pseudotyped virus assay. Compound **8a** was implicated to block the viral entry through inhibition of cathepsin B, even though there was no direct binding between **8a** and cathepsin B [26]. Compound **8a** was inactive against cathepsin L, which is an endosomal protease involves in SARS-CoV-2 entry through endocytosis. To test whether our compounds inhibit cathepsin B or cathepsin L, the enzyme inhibitory activity of compound **5** was determined using cathepsin B or cathepsin L inhibitor assay kits (BPS bioscience). Compound **5** was totally inactive against cathepsin B or cathepsin L at a concentration as high as 20 µM whereas the known cathepsin B inhibitor E64 inhibited the enzyme activity by more than 95% at a concentration as low as 0.1 µM (Figure 2). The result strongly suggests that cathepsin B, or cathepsin L, is not a direct target of compound **5**. Thus, the molecular mechanism of action of the aloperine derivatives remains to be determined.

## 3. Discussion

Although there are two antiviral drugs currently available to treat COVID-19, there is no approved small molecule drug that targets SARS-CoV-2 entry. With the high mutation rate of the virus, variants that resistant to the antiviral drugs are likely to emerge in the future. Thus, more drug candidates are urgently needed for antiviral drug development for COVID-19 treatment, especially those target different steps of viral replication cycle. Effective small molecule entry inhibitors may have potential to become a useful addition to current therapy against SARS-CoV-2 infection.

The SARS-CoV-2 spike-pseudotyped virus system has provided a convenient method to screen compound libraries for potential hits that block SARS-CoV-2 entry [23,26,27,28,29]. Many of the positive hits were showing moderate potency. Aloperine is a natural product isolated from *Sophora alopecuroides* L. and other plant species [30,31]. It has been tested in cell and animal models for its potential therapeutic effects, such as regulation of inflammatory cytokines [32,33]. We have previously shown that aloperine exhibited moderate inhibitory activities against HIV and influenza A viruses [19,20,21,22]. In this study, we demonstrated that aloperine inhibited the SARS-CoV-2 entry with a moderate IC_50_ of 11.5 µM. However, with a simple structural modification at the N12 position, the aloperine derivative compound **5** was transformed into a much potent compound with an approximately 22-fold improvement in potency against the viral entry. It should be noted that all the tested Omicron variants, including the currently circulating BA.4/BA.5, were sensitive to compound **5**. This result is consistent with the notion that compound **5** may not interfere with receptor binding of the spike proteins, as the receptor-binding domain of BA.4/BA.5 is significantly different from that of D614G variant in primary amino acid sequence and receptor affinity [13].

A recent report showed that an aloperine derivative (**8a**) could block SARS-CoV-2 entry through inhibition of cathepsin B at an IC_50_ of 19.1 µM in a similar pseudotype virus assay [26]. In contrast, compound **5** did not inhibit the activity of cathepsin B or cathepsin L, and was able to inhibit SARS-CoV-2 and its variants at sub-micromolar concentrations. The differences in potency and possible mechanisms of action between the two compounds are likely due to their respective N12 side chains. The N12 side chain (CH_2_)_4_NHCH_2_Ph of compound **5** (Table 1) is longer and possesses higher chemical complexity when compared with that on compound **8a** (*p*-ClPh(CH_2_)_2_). We have previously shown that aloperine may function as a privileged scaffold and N12 side chain modifications result in derivatives with different biological activities, such as specific anti-HIV activity or anti-influenza A virus activity [19,20,21,22]. Therefore, it is possible that minor change in the N12 side chain of aloperine could result in differences in mechanisms of action. Identification of the direct target(s) of the compounds would provide a more definitive molecular detail on how the compounds inhibit SARS-CoV-2 entry.

## 4. Materials and Methods

### 4.1. Materials

293T-ACE2 cells were kindly provided by Integral Molecular, PA and maintained in DMEM supplements with 10% FBS and 0.5 µg/mL of puromycin. Monoclonal antibody DH1047, and spike-pseudotyped viruses correlated to D614G, Delta, Omicrons including BA.1, BA.2, and BA.4/BA.5 were kindly provided by Dr. David Montefiori of Duke University. Aloperine was purchased from Sigma-Aldrich (Cat. # 546704, St. Louis, MO, USA). E64D was purchased from Selleck Chemicals (Cat. # S7393, Houston, TX, USA).

### 4.2. SARS-CoV-2 Pseudovirus Inhibition Assay

The anti-SARS-CoV-2 activity of the aloperine derivatives was assessed with a D614G or other above mentioned variant spike-pseudotyped viruses in 293T-ACE2 cells as a function of reductions in luciferase (Luc) reporter activity as described by D. Weissman et al. previously [23]. The assay system was kindly provided by Dr. David Montefiori at Duke University. Briefly, pseudovirions were produced by FuGENE^®^ 6 (Promega, Madison, WI, USA, Cat. # E2691) transfection of HEK293T cells with a plasmid mixture containing a spike protein plasmid (VRC7480.D614G), a lentiviral backbone plasmid (pCMV Δ8.2), and a firefly Luc reporter gene plasmid (pHR’ CMV *Luc*), which produces luciferase upon successful viral infection) in a 1:17:17 ratio. Pseudovirus in culture medium was collected after an additional 2 days of incubation. The murine leukemia virus envelope protein (MLV-Env) pseudotyped virus was produced in the same assay system except that the spike protein plasmid (VRC7480.D614G) was replaced with pSV-A-MLV-env (NIH AIDS Reagent Program, ARP1065). For virus entry inhibition, the D614G spike-pseudotyped virus and 293T-ACE2 cells were treated with various concentrations of compounds for 3 days. Luminescence was then measured by adding the Promega Bright-Glo luciferase reagent and using a PerkinElmer Victor 2 luminometer. IC_50_ was calculated as compound concentration at which relative luminescence units (RLU) were reduced by 50% compared to virus control wells after subtraction of background RLUs.

### 4.3. Immunofluorescence Staining of the SARS-CoV-2 Spike Proteins and Confocal Microscopy

293T-ACE2 cells cultured in 96-well glass-bottom plates were treated with compound **5** (5 µM) and infected with D614G spike-pseudotyped virus for 2 h. The cells were fixed with 4% formaldehyde in PBS for 15 min. The cells were then treated with a blocking buffer containing 5% FBS and 0.3% Triton X-100 in PBS for 60 min. Immunostaining was carried out by incubating Alexa Fluor 488-conjugated anti-SARS-CoV-2 spike protein antibody (Thermo Fisher Scientific, Waltham, WA, USA, Cat. # 53-6490-82) with the cells at 4 °C overnight. The samples were then washed three times in PBS before treated with Prolong^®^ Gold Anti-Fade Reagent with DAPI (Cell Signaling Technology, Danvers, MA, USA, Cat. # 4083S). Confocal images were acquired using a Nikon TE2000-U laser-scanning confocal microscope (Nikon, Tokyo, Japan). Confocal image analysis was performed with NIS-Elements AR 3.0 software (Nikon).

### 4.4. Cytotoxicity Assay

The CellTiter-Glo^®^ Luminescent Cell Viability Assay (Promega, Cat. # G7570 ) was used to determine the cytotoxicity of the aloperine derivatives. 293T-ACE2 cells were cultured in the presence of various concentrations of the compounds for 3 days. The cytotoxicity of the compounds was determined by following the protocol provided by the manufacturer. The 50% cytotoxic concentration (CC_50_) was defined as the concentration that caused a 50% reduction in cell viability.

## 5. Conclusions

In summary, the results of this study are consistent with the notion that aloperine is a privileged scaffold that can be structurally optimized to have selective antiviral activity. Spike protein-pseudotyped viruses of major SARS-CoV-2 subtypes such as D614G, Delta, Omicron BA.1, BA.2, and BA.4/BA.5, were included to evaluate the spectrum and potency of compound **5**. Our results indicated that compound **5** could inhibit all the tested pseudotyped viruses at sub-µM concentrations. We proposed a model to suggest that compound **5** may inhibit SARS-CoV-2 infection through inhibition of viral entry at membrane fusion and/or endocytosis pathways after the virus binding to receptors (Figure 3). We speculate that the viral particles were arrested at a stage before the viral fusion with cellular and/or endosomal membranes based on the fluorescent puncta observed from compound **5**-treated 293T-ACE2 cells under confocal microscopy (Figure 1). However, the molecular details of how compound **5** arrests the viral entry remain to be determined. Nevertheless, the submicromolar anti-SARS-CoV-2 entry activity of compound **5** in all tested variants offers a promising lead for further developing aloperine derivatives as anti-COVID-19 drug candidates.

## Figures and Tables

**Figure 1 ijms-23-09659-f001:**
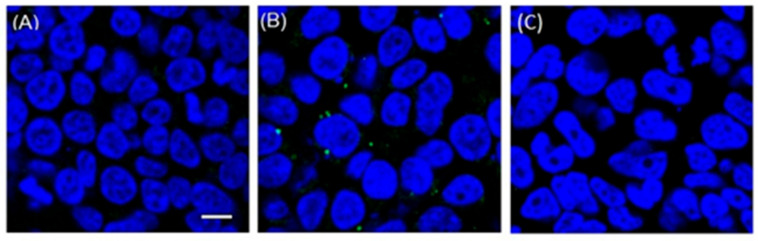
Compound **5** arrests D614G spike-pseudotyped virus entry. D614G spike-pseudotyped viruses were used to infect 293T-ACE2 cells, which were stained with DAPI (nuclear stain, blue) 2 h post infection. Size bar = 10 µm. Confocal microscopy images were acquired using a Nikon A1R confocal microscope with a 60×/1.4 NA oil-immersion Plan-Apochromat lens. (**A**) 293T-ACE2 cells were infected with D614G spike-pseudotyped virus; (**B**) experiment was performed in the presence of compound **5** at 5 μM; (**C**) experiment was performed in the presence of chloroquine diphosphate at 5 μM.

**Figure 2 ijms-23-09659-f002:**
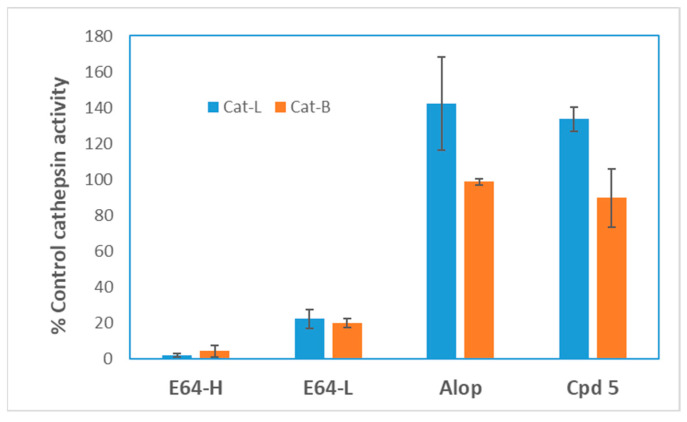
Aloperine derivatives were inactive against cathepsin B and L. Compound **5** and aloperine were tested for their inhibitory activity against cathepsin B or L using a BPS bioscience assay kit and the protocol provided by the manufacturer (catlog#79590). The enzyme activity in the absence of compounds (control) was defined as 100%. Aloperine (Alop) and compound **5** (Cpd 5) were tested at 20 µM. The known cathepsin B inhibitor E64 was tested at 0.1 µM (E64-H) and 0.01 µM (E64-L), respectively. The data represent the average of a duplicated experiment.

**Figure 3 ijms-23-09659-f003:**
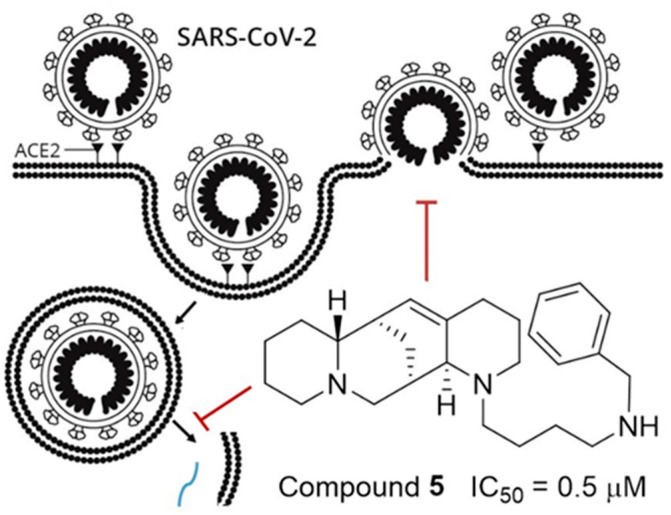
Model for inhibition of SARS-Cov-2 entry by Compound **5**.

**Table 1 ijms-23-09659-t001:** Inhibition of D614G spike-pseudotyped virus entry by aloperine derivatives.

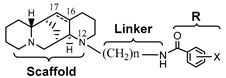	D614G Spike-PseudotypedVirus	293T-ACE2	PA/Puerto Rico/8/1934Influenza A Virus (PR8)	HIV-1_NL4-3-nanoluc-sec_
Cpds	Linker (n)	R	IC_50_ ^1^	CC_50_ ^2^	IC_50_ ^1^	IC_50_ ^1^
Alop ^3–6^	none	none	11.5 ± 2.3	>20	14.5	1.75
1 (*cis*) ^6,7^	6	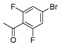	4.7 ± 0.76	>20	0.091	>20
2 ^3,6^	2	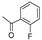	Inactive ^8^	ND ^9^	5.5	>25
3 ^5,6^	4	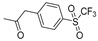	3.8 ± 0.68	>20	>20	0.12
4 ^4^	5	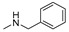	0.86 ± 1.5	>20	>40	0.84
5 ^4^	4	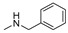	0.5 ± 0.12>20 **	>20	>40	0.96
6 ^4,6^	4	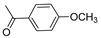	11.9 ± 2.1	>20	>20	11.4
7 ^6^	6	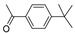	3.7 ± 0.87	>20	0.83	0.80
8 ^6^	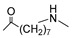	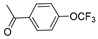	17.5 ± 2.3	>20	>28.5	>20
9	4	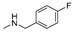	0.61 ± 0.18	>10	4.8 ± 1.4	Inac ^8^
Chloroquine diphosphate	2.3 ± 3.6	ND ^9^	ND ^9^	ND ^9^

^1^ Concentration (µM) required to inhibit an indicator virus, D614G spike-pseudotyped virus (D614G variant), PA/Puerto Rico/8/1934 (PR8), or NL4-3-nanoluc-sec infection by 50%. ^2^ Concentration (µM) that reduced the viability of 293T-ACE2 cells by 50%. ^3−6^ The compound and its anti-HIV and/or anti-influenza virus activity were previously described in part in references [19,20,21,22], respectively. ^7^ *cis* denotes that double bond of the quinolizidine scaffold was reduced and the compound is in the *cis*-conformations. ^8^ CC_50_/IC_50_ ratio < 5. ^9^ not determined. The IC_50_ values on D614G spike-pseudotyped virus are presented as the mean ± SD of three tests. ** denotes the antiviral activity against the murine leukemia virus envelope (MLV-env) pseudotyped virus.

**Table 2 ijms-23-09659-t002:** Effects of compound **5** against SARS-CoV-2 variants (spike-pseudotyped viruses).

Variants	IC_50_ ^1^		
	Cpd 5 (µM)	E64D (µM)	DH1047 (µg/mL)
D614G	0.53 ± 0.12	0.48 ± 0.11	1.2 ± 2.1
Delta	0.58 ± 0.15	0.45 ± 0.082	0.72 ± 0.12
Omicron BA.1	0.76 ± 0.22	0.36 ± 0.076	>10
Omicron BA.2	0.83 ± 0.21	0.37 ± 0.066	>10
Omicron BA.4/BA.5	0.86 ± 0.25	0.34 ± 0.081	>10

^1^ Concentration required to inhibit variant spike-pseudotyped virus infection by 50%. The IC_50_ values of compound **5** on each of the variant spike-pseudotyped viruses are presented as the mean ± SD of three tests.

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
