# Peer review of "Quinolizidines as Novel SARS-CoV-2 Entry Inhibitors"

_ijms, 2022, doi:10.3390/ijms23179659_

Round 1
Reviewer 1 Report
This manuscript reports the discovery of collection of aloperine derivatives that inhibit entry of viral particles pseudotyped with SARS-CoV-2 spike proteins corresponding to the G614, Delta or Omicron variants. Agents of the same chemical class, but with different S.A.R., were previously reported as inhibitors of HIV and influenza virus infections.
The discovery of anti-coronavirus activity of aloperine derivatives is novel and potentially interesting. However, there are several issues that make the present paper not suitable for publications (listed below):
1) The authors do not specify what type of pseudotype viral particle they used for the experiments. Lentiviral particles? VSV? In any case, it would be important to show a specificity control, i.e., that the aloperine derivatives do not inhibit entry of similar particles pseudotyped with a different envelop (e.g., VSV glycoprotein). In the absence of such a control experiment, the authors cannot claim that inhibition is specific for SARS-CoV-2 spike.
2) If possible, it would be best to also show inhibition of infection with authentic SARS-CoV-2.
3) Page 5, lines 154-155. “The SARS-CoV-2 can enter the cells and subsequently release and disperse the viral spike protein in the cytosol”. Where does this notion come from? The authors should provide a reference.
4) The data of Figure 1 are grossly overinterpreted. First of all, there is no size bar on the micrographs and it is not specified what lenses were used for the acquisition. Second, conclusions cannot be drawn based on comparison of single fields. All I see is no Spike signal in panel A and punctuate staining in panel B. Is this difference statistically significant? Multiple fields from independent experiments need to be considered, analyzed quantitatively and evaluated for statistical significance. In any case, the interpretation provided by the authors (i.e., compound 5 inhibits fusion and not binding) is far too stretched. Such a conclusion can hardly be reached from microscopy data alone (see also below).
5) In order for the authors to show that binding of spike to the receptor is not inhibited, they need to evaluate compound(s) in a ACE2-Spike binding assay. In order for the author to show that fusion is inhibited, they need to evaluate compound(s) in a fusion assay. In the recent literature, there is plenty of examples of how such assays could be performed.
6) Figure 2 does not have error bars. The legend says these data are “average of a duplicate experiments”. Were these independent experiments of just technical duplicates? From the graph the reader cannot appreciate the reproducibility/variability associated with these data.
In summary, the manuscript reports potentially interesting novel antiviral molecules, but the authors fail to demonstrate that what they observe is specific for SARS-CoV-2 and that antiviral activity is present on authentic virus. Moreover, they do not show any sensible data supporting the mechanism of action they propose. In my opinion, the paper, in its present form, is not acceptable for publication
Reviewer 2 Report
The manuscript by Huang et al. “Quinolizidines as Novel SARS-CoV-2 Entry Inhibitors” demonstrated that Compound 5 is a potent inhibitor of SARS-CoV-2 along with its other variants. Overall, the manuscript is well-demonstrated and interesting. It requires revision as follows:
Comments
1. The author should prepare an illustration of the mechanism of compound (5) action towards SARS-CoV-2 and its variants.
2. In lines 43-45, the information can be more elaborated in details such as i) examples and severity of its variants; ii) pandemic threats by their continuous evolution via the emergence of hybrid variants, and the possibility of more waves in near future, and iii) vaccines limitations and future preventive measures i.e. doi:10.1007/s15010-021-01734-2; doi:10.1016/j.ijsu.2022.106727. Such points also can be used for the improvement of the discussion section (minor).
3. Introduction, please clearly specify the main objective of the present study, and its significance.
4. Please add a conclusion section.
Reviewer 3 Report
In this report, Li Huang & al have tested against SARS-CoV-2 pseudotyped lentiviruses a series of 8 aloperine derivatives that are known for blocking the entry step of HIV or influenza virus. This led to the design, synthesis and evaluation of a novel aloperine derivative (Compound 9) that inhibits SARS-CoV-2 entry step with an IC50 ranging from 0.5 to 0.8 microM depending on the spike variant. Unfortunately, the impact of this manuscript is limited by several shortcomings.
Major points:
- Aloperine derivatives were previously described for inhibiting SARS-CoV-2 infection (PMID: 34333425). Although compound 9 is significantly more active than compound 8a described by Kun Wang & al., the current report by Li Huang & al. is more an increment of a previous report than a new finding.
- Compounds were only tested on SARS-CoV-2 pseudotyped particles. The antiviral activity must be validated on SARS-CoV-2. Furthermore, this should be performed in a more relevant cellular model (human lung epithelial cells; cells expressing TMPRSS2).
- Understanding the mode of action of compound 9 is essential to evaluate the potential of this molecule. Many molecules inhibiting SARS-CoV-2 entry, including hydroxychloroquine, are cationic amphiphilic drugs (CADs) with lysosome alkalizing properties and promoting phospholipidosis. These drugs are now known to be inactive in vivo (PMID: 34326236). Based on its structure, it is possible that Compound 9 falls in this category. It is critical to address this question to evaluate the interest of Compound 9 as a SARS-CoV-2 inhibitor.
- It is shown that Compound 9 is not a direct inhibitor of Cathepsin B (Figure 2) but leads to the accumulation of the Spike protein into intracellular compartments (Figure 1). Again, this strongly suggests that Compound 9 prevents the degradation of viral proteins by interfering with lysosomes. This is the kind of phenotype that you would expect for a cationic amphiphilic drug interfering with lysosomal proteases by increasing the pH of this acidic compartment.
- Standard deviation and statistics are missing in Figure 2.
Minor points:
There is some level of redundancy in Page 3 lines 80-85 and 87-91. Please avoid back-and-forth with previous data on HIV and influenza virus.
Page 7, line 244. The indicated reference 18 does not correspond to Weismann et al.
The Material and Methods section is brief and information is missing (cell culture medium, cell concentrations per well/volume, more details about pseudotyped lentiviruses should be provided even if there is a reference to a previous report)
Round 2
Reviewer 2 Report
The authors have partially revised the manuscript.
Comments
1. The author should revise Graph 1 in the detailed description (text labeling) for better presentation.
2. In lines 45-49, the information can be more elaborated as per previous version manuscript suggestions (Comment 2) including recent COVID-19 variants.
3. Please add the significance of the present study at the end of the Introduction.
4. Improve discussion (minor) based on the issues of recent COVID-19 variants.
